# Antiviral and Anti-Inflammatory Activities of Fluoxetine in a SARS-CoV-2 Infection Mouse Model

**DOI:** 10.3390/ijms232113623

**Published:** 2022-11-07

**Authors:** David Péricat, Stephen Adonai Leon-Icaza, Marina Sanchez Rico, Christiane Mühle, Iulia Zoicas, Fabian Schumacher, Rémi Planès, Raoul Mazars, Germain Gros, Alexander Carpinteiro, Katrin Anne Becker, Jacques Izopet, Nathalie Strub-Wourgaft, Peter Sjö, Olivier Neyrolles, Burkhard Kleuser, Frédéric Limosin, Erich Gulbins, Johannes Kornhuber, Etienne Meunier, Nicolas Hoertel, Céline Cougoule

**Affiliations:** 1Institute of Pharmacology and Structural Biology (IPBS), University of Toulouse, CNRS, 31000 Toulouse, France; 2Faculté de Santé, Université Paris Cité, 75006 Paris, France; 3Département de Psychiatrie et d’Addictologie de l’Adulte et du Sujet Agé, Assistance Publique-Hôpitaux de Paris (AP-HP), DMU Psychiatrie et Addictologie, Hôpital Corentin-Celton, 92130 Issy-les-Moulineaux, France; 4INSERM, Institut de Psychiatrie et Neurosciences de Paris (IPNP), UMR_S1266, 75014 Paris, France; 5Department of Psychiatry and Psychotherapy, University Hospital, Friedrich-Alexander-University of Erlangen-Nuremberg, 91054 Erlangen, Germany; 6Institute of Pharmacy, Freie Universität Berlin, Königin-Luise-Str. 2-4, 14195 Berlin, Germany; 7Institute for Molecular Biology, University Medicine Essen, University of Duisburg-Essen, 47057 Essen, Germany; 8Toulouse Institute for Infectious and Inflammatory Diseases (INFINITy), Université Toulouse, CNRS, INSERM, UPS, 31300 Toulouse, France; 9Laboratoire de Virologie, CHU Toulouse, Hôpital Purpan, 31300 Toulouse, France; 10Drugs for Neglected Diseases Initiative, 1202 Geneva, Switzerland

**Keywords:** COVID-19, SARS-CoV-2, mouse model, fluoxetine, anti-depressant, inflammation

## Abstract

The coronavirus disease 2019 (COVID-19) pandemic continues to cause significant morbidity and mortality worldwide. Since a large portion of the world’s population is currently unvaccinated or incompletely vaccinated and has limited access to approved treatments against COVID-19, there is an urgent need to continue research on treatment options, especially those at low cost and which are immediately available to patients, particularly in low- and middle-income countries. Prior in vitro and observational studies have shown that fluoxetine, possibly through its inhibitory effect on the acid sphingomyelinase/ceramide system, could be a promising antiviral and anti-inflammatory treatment against COVID-19. In this report, we evaluated the potential antiviral and anti-inflammatory activities of fluoxetine in a K18-hACE2 mouse model of SARS-CoV-2 infection, and against variants of concern in vitro, i.e., SARS-CoV-2 ancestral strain, Alpha B.1.1.7, Gamma P1, Delta B1.617 and Omicron BA.5. Fluoxetine, administrated after SARS-CoV-2 infection, significantly reduced lung tissue viral titres and expression of several inflammatory markers (i.e., IL-6, TNFα, CCL2 and CXCL10). It also inhibited the replication of all variants of concern in vitro. A modulation of the ceramide system in the lung tissues, as reflected by the increase in the ratio HexCer 16:0/Cer 16:0 in fluoxetine-treated mice, may contribute to explain these effects. Our findings demonstrate the antiviral and anti-inflammatory properties of fluoxetine in a K18-hACE2 mouse model of SARS-CoV-2 infection, and its in vitro antiviral activity against variants of concern, establishing fluoxetine as a very promising candidate for the prevention and treatment of SARS-CoV-2 infection and disease pathogenesis.

## 1. Introduction

The SARS-CoV-2 pandemic has created an economic and health crisis worldwide [1,2,3,4]. Since a large proportion of the world’s population is currently unvaccinated, or incompletely vaccinated, the availability of outpatient treatment options for COVID-19, such as nirmatrelvir-ritonavir (Paxlovid^®^), neutralizing monoclonal antibodies, remdesivir, and molnupiravir, has raised hopes for reducing COVID-19-related morbidity and mortality. However, the use of these treatments may be held back due to either their medical indications (e.g., within 5 days of symptom onset for nirmatrelvir-ritonavir, specific variants for sotrovimab), contraindications (e.g., concomitant use of drugs that are potent CYP3A inducers, severe hepatic impairment or severe renal impairment for nirmatrelvir-ritonavir [5]), the need of intravenous therapy (for neutralizing monoclonal antibodies and remdesivir), limited supplies, and substantial cost [6]. This supports the urgent need for research on other outpatient treatment options, especially those at lower cost and which are immediately available to allow their use, particularly in low- and middle-income countries with limited access to vaccines and approved treatments against COVID-19 [7,8,9].

Prior research suggests that certain selective serotonin reuptake inhibitor (SSRI) antidepressants, particularly fluoxetine and fluvoxamine, could be beneficial against COVID-19 [6,10,11], and thus may be a potential means of reaching this goal. First, several preclinical studies demonstrated the in vitro efficacy of fluoxetine against different variants of SARS-CoV-2 in human and non-human (i.e., Vero E6, Calu-1, Calu-3, HEK293T-ACE2-TMPRSS2) cells [12,13,14,15,16,17,18,19]. Second, five retrospective observational cohort studies [20,21,22,23,24] of patients with COVID-19 in the acute-care setting reported reduced death or mechanical ventilation in those prescribed with SSRIs prior to hospitalization—particularly fluoxetine. Another observational study [25] also found that exposure to antidepressants, especially those with functional inhibition of acid sphingomyelinase (FIASMA) properties such as fluoxetine, was associated with reduced incidence of emergency department visitation or hospital admission among SARS-CoV-2 positive outpatients, in a dose-dependent manner and from daily doses of at least 20 mg fluoxetine equivalents. Moreover, a retrospective cohort study conducted in an adult psychiatric facility suggested a significant negative association of prior antidepressant use—particularly fluoxetine—with laboratory-detectable SARS-CoV-2 infection [26], a finding that has been recently replicated in another observational study [24]. Third, in the ambulatory setting, two prospective randomized, placebo-controlled trials [27,28,29] and one non-randomized open-label clinical study [30], as well as a meta-analysis [31], found a significant association between the short-term use (10–15 days) of fluvoxamine prescribed at a dose between 100 and 300 mg/d within 7 days of symptom onset and reduced risk of clinical deterioration. Contrariwise, an RCT of low-dose fluvoxamine [32,32] (i.e., 100 mg/d) among overweight or obese outpatients with COVID-19 showed no significant benefit on emergency department visits, hospitalizations or death. Finally, a prospective cohort study [33] of patients admitted to intensive care unit for COVID-19 reported a significant association between the 15-day use of fluvoxamine prescribed at 300 mg/d and reduced mortality. Taken together, these findings support that certain antidepressants, and particularly fluoxetine or fluvoxamine, could be efficient when prescribed at doses of at least 20–40 mg fluoxetine equivalents [24,25] in limiting clinical deterioration of patients infected with SARS-CoV-2 in both ambulatory and acute-care settings.

Although the mechanisms underlying this potential therapeutic effect of fluoxetine and fluvoxamine against COVID-19 are likely to be multiple and interrelated, several studies support that the interaction with the acid sphingomyelinase (ASM)/ceramide system, resulting in both antiviral and anti-inflammatory effects, may be a prominent one [13]. First, these SSRI antidepressants belong to the group of functional inhibitors of acid sphingomyelinase (FIASMA) [13,34]. In vitro and in vivo, these pharmacological compounds inhibit ASM, an enzyme that catalyses the hydrolysis of sphingomyelin to ceramide and phosphorylcholine, by detaching it from the lysosomal membrane and thereby triggering its degradation [34]. Preclinical data indicate that SARS-CoV-2 activates the ASM/ceramide system, resulting in the formation of ceramide-enriched membrane domains, which facilitate viral entry and infection by clustering ACE2, the cellular receptor of SARS-CoV-2, and the release of pro-inflammatory cytokines [12,13,35]. Importantly, the reconstitution of ceramides in cells treated with FIASMA antidepressants restored the infection [12]. In healthy volunteers, oral use of the FIASMA antidepressant amitriptyline prevented infection of freshly isolated nasal epithelial cells, which is also restored after the reconstitution of ceramides in these cells [12]. In observational retrospective studies, the use of a FIASMA medication upon hospital admission was significantly associated with reduced mortality or mechanical ventilation [36]. Finally, plasma levels of certain sphingolipids, including increased levels of ceramides, decreased levels of their derivatives hexosylceramides, and decreased hexosylceramide/ceramide ratio, were found to correlate with clinical disease severity and inflammation markers [37,38,39] in patients with COVID-19. Altogether, these data support the central role of the ASM/ceramide system in SARS-CoV-2 infection and the potential of FIASMA antidepressants in prevention and treatment of COVID-19.

Among SSRIs, the magnitude of the in vitro inhibition of ASM appears to correlate with the magnitude of the in vitro antiviral effect against SARS-CoV-2 [13,15,40] (e.g., amitriptyline > fluoxetine > paroxetine > fluvoxamine > citalopram). While amitriptyline exhibits potential toxicity issues in elderly, fluoxetine displays the highest FIASMA activity. In addition, fluoxetine shows anti-inflammatory properties in lipopolysaccharide (LPS)-challenged mouse models [41,42], which may be of interest in patients infected with SARS-CoV-2, as increased levels of proinflammatory cytokines in serum (e.g., IL1B, IL6, IFNγ, CXCL10, and CCL2) are associated with pulmonary inflammation and extensive lung damage [43].

While observational and in vitro data on fluoxetine are promising, experiments examining its effect against SARS-CoV-2 infection in an animal model are lacking. Mice are not naturally permissive to SARS-CoV-2 infection, due to the low affinity of the S protein receptor binding domain for mouse ACE2. Previously, a transgenic mouse expressing the human ACE2 gene under the control of the cytokeratin 18 (K18) promoter (the K18-hACE2 mouse) was generated to study SARS-CoV pathogenesis [44]. Recent studies have demonstrated that K18-hACE2 mice support SARS-CoV-2 infection and develop a lethal respiratory illness with weight loss and inflammation [45,46,47].

In this report, we tested the potential antiviral and anti-inflammatory activities of fluoxetine against SARS-CoV-2 in a K18-hACE2 mouse model of infection, and against several variants of concern in vitro, and tested the hypothesis of the implication of ceramides and/or their derivatives hexosylceramides.

## 2. Results

### 2.1. Antiviral Activity

Viral loads were evaluated from 44 K18-hACE2 mice euthanized at day 2 and day 6 post-infection. As shown in Figure 1B, fluoxetine treatment significantly reduced the viral load in the lung of SARS-CoV-2 infected mice at both day 2 (log_10_ TCID_50_/mL 6.19 (±0.14) vs. 4.04 (±0.44); Mann-Whitney test *p* = 0.0012) and day 6 (log_10_ TCID_50_/mL 4.22 log_10_ (±0.19) vs. 3.28 (±0.17); *p* = 0.0048). Moreover, viral RNA in lung homogenates was significantly reduced at day 2 post-infection in fluoxetine-treated mice compared to vehicle-treated mice (9.91 × 10^4^ (± 4.15 × 10^4^) vs. 1.04 × 10^6^ (± 6.60 × 10^5^); *p* = 0.0310) (Figure 1C). Viral RNA in lung homogenates was still significantly reduced at day 2 post-infection in fluoxetine-treated mice compared to vehicle-treated mice when excluding the 2 outliers in the saline-treated group (3.85 × 10^4^ (± 2.50 × 10^4^) vs. 3.85 × 10^5^ (± 1.22 × 10^5^); *p* = 0.0295) (Appendix A). These results show that fluoxetine displays a significant and substantial antiviral activity against SARS-CoV-2 in mice.

As shown in Figure 1D, fluoxetine significantly reduced expression of IL-6 in the lung tissues at both day 2 (IL-6 8.97 × 10^−4^ (± 4.09 × 10^−4^) vs. IL-6 1.06 × 10^−1^ (± 4.88 × 10^−2^); *p* = 0.0017) and day 6 post-infection (IL-6 3.01 × 10^−3^ (± 1.58 × 10^−3^) vs. IL-6 2.31 × 10^−3^ (±6.11 × 10^−3^); *p* = 0.0140), and other pro-inflammatory cytokines at day 2 post-infection, including TNFα (8.07 × 10^−4^ (± 1.21 × 10^−4^) vs. 2.24 × 10^−3^ (± 3.70 × 10^−4^); *p* = 0.0004), CXCL10 (2.44 × 10^−3^ (± 7.17 × 10^−4^) vs. 9.99 × 10^−2^ (± 4.14 × 10^−2^); *p* < 0.0001) and CCL2 (8.83 × 10^−4^ (± 1.21 × 10^−4^) vs. 2.24 × 10^−3^ (± 3.70 × 10^−4^); *p* < 0.0001). A reduction in pro-inflammatory cytokine expression did not appear to be solely related to reduced viral load in fluoxetine-treated mice. Indeed, the difference in levels of IL-6 and TNFα between fluoxetine-treated and vehicle-treated mice was still significant after adjusting successively for viral load and viral RNA at both day 2 and day 6 (Appendix A). In addition, IL-6 expression level did not correlate with lung viral load or RNA load in both fluoxetine-treated (lung viral load, r = −0.11, *p* = 0.7131; lung viral RNA load, r = −0.18, *p* = 0.5403) and vehicle-treated mice (lung viral load, r = −0.14, *p* = 0.6222; lung viral RNA load, r = −0.22, *p* = 0.4508) (Appendix A). These results were maintained when excluding three outliers (one fluoxetine- and one vehicle-treated mouse at day 2, and one fluoxetine-treated mouse at day 6) from the analyses (Appendix A). To evaluate the immunomodulatory property of fluoxetine, independently of the viral load, we used the A549-Dual™ cells consisting in a human NF-κB-SEAP & IRF-Luc Reporter lung carcinoma cell line. Upon cell stimulation with PolyI:C or TNFa, fluoxetine inhibited NF-κB activation, while IRF pathway was not affected (Appendix A), indicating that fluoxetine displays, as previously described [41], an immunomodulatory function on the NF-κB pathway. Altogether, these findings suggest that fluoxetine dampens inflammation during SARS-CoV-2 infection, independently of its antiviral properties.

### 2.2. Body Weight and Temperature

While fluoxetine at 40 mg/kg is well tolerated over a period of 15 days [48], it induces body weight loss in the first 24 h of treatment [49]. We showed that fluoxetine induced a significant body weight loss after the first 24 h in K18-hACE2 mice, while the body weight was constant in vehicle-treated mice (Appendix A). Moreover, we confirmed that vehicle-treated K18-hACE2 mice started to lose weight 2 days post-infection [46], while the body weight was stable longer in fluoxetine-treated mice (Appendix A). Further comparison of body weight between groups was biased due to the initial body weight loss in fluoxetine-treated mice, and did not show an overall significant difference (Appendix A). While body temperature was reduced at day 6 in vehicle-treated K18-hACE2 mice (compared to non-infected mice (37 ± 0.3 °C, *n* = 4)) as a sign of critical illness [50], we measured a significantly improved body temperature in fluoxetine-treated mice (33.3 ± 1.00 °C vs. 28.6 ± 0.94 °C; *p* = 0.0052) (Appendix A).

### 2.3. Fluoxetine Modulates the Ceramide System

We evaluated whether fluoxetine could modulate the ceramide system in the lung tissues of mice infected with SARS-CoV-2. As shown in Figure 2B,C, fluoxetine treatment significantly increased levels of hexosylceramides 16:0 (166.93 (± 22.4) vs. 101.65 (±11.0); *p* = 0.0186) and the ratio HexCer 16:0/Cer 16:0 (0.40 (±0.05) vs. 0.28 (±0.03); *p* = 0.0408). There were no significant between-group differences in ceramides 16:0 and sphingomyelin 16:0 levels (Figure 2A,D). The between-group differences in ceramides 16:0 and sphingomyelin 16:0 remained non-significant when excluding outliers from the analyses (one fluoxetine-treated mouse in ceramides 16:0, and one fluoxetine-and one saline-treated mouse in sphingomyelin 16:0) (Appendix A).

### 2.4. Anti-Viral Activity of Fluoxetine In Vitro against Variants of Concern

Vero E6 cells were successively infected with Alpha B.1.1.7, Gamma P1, Delta B1.617 and Omicron BA.5 variants of concern, and viral cytopathic effect and replication were measured. As shown in Figure 3A, fluoxetine, at the non-cytotoxic dose of 12.5 µM (Appendix A) inhibited replication of all the variants of concern tested. Finally, we evaluated fluoxetine effectiveness in the human A549 lung carcinoma cells stably overexpressing the SARS-CoV-2 receptor ACE2, and the protease TMPRSS2 to increase their permissiveness to SARS-CoV-2 infection [51]. A549-hACE2-TMPRSS2 cells were permissive to infection by all variants of concern (Figure 3B). A549-hACE2-TMPRSS2 cells infected with SARS-CoV-2 ancestral strain, Alpha B.1.1.7, Gamma P1, Delta B1.617 and Omicron BA.5 variants of concern, and treated with fluoxetine (12.5 µM) showed reduced amount of infectious virus and restored cell viability (Figure 3B). Altogether, these results show that fluoxetine is efficient at inhibiting replication across several SARS-CoV-2 variants of concern.

### 2.5. Anti-Viral Activity of Fluoxetine Ex Vivo on Human Primary Airway Epithelial Cells

Human airway organoid-derived epithelia cultured in 2D were infected with SARS-CoV-2 as previously described [52], and the percentage of infection was microscopically quantified (Figure 4A). We confirmed that SARS-CoV-2 mainly infects ciliated cells revealed by the AcTUB staining. As shown in Figure 4B, fluoxetine significantly inhibited infection of human primary airway epithelial cells in a dose-dependent manner.

## 3. Discussion

In this report, we show that fluoxetine has both significant anti-viral and anti-inflammatory effects in a K18-hACE2 mouse model of SARS-CoV-2 infection, both mechanisms appearing to be potentially, at least partly, independent to each other. We also show that fluoxetine efficiently inhibits the replication of different SARS-CoV-2 variants of concern, including the SARS-CoV-2 ancestral strain, Alpha B.1.1.7, Gamma P1, Delta B1.617 and Omicron BA.5.

These findings support prior preclinical studies that showed an in vitro efficacy of fluoxetine against SARS-CoV-2 in different host cells (i.e., Vero E6, Calu-1, Calu-3, HEK293T-ACE2-TMPRSS2 cells) and human lung epithelial cells [12,17,18]. They are also in line with results of observational studies showing significant associations between fluoxetine use and reduced COVID-19-related mortality or mechanical ventilation [20,22,24,25] and laboratory-detectable SARS-CoV-2 infection [24,26], and extend them by indicating that fluoxetine effects could be observed quickly when administrated after the start of the infection. Furthermore, they are consistent with results from animal models of septic shock and allergic asthma showing that fluoxetine markedly reduces the inflammatory reaction [42]. Finally, these results are in line with clinical studies conducted among individuals with major depressive disorder, showing that several antidepressants, including fluoxetine, significantly decrease peripheral levels of IL-6, TNF-α, IL-10, and CCL-2 [53], which have been shown to increase in severe COVID-19 patients [54].

By showing a significant increase in lung levels of hexosylceramides 16:0 and in the ratio HexCer 16:0/Cer 16:0 in fluoxetine versus vehicle-treated mice, our results suggest that the ceramide system might be implied in the observed antiviral and/or anti-inflammatory effects. Further analysis is now required to evaluate overtime, in comparison to baseline levels, those lipids dynamics during SARS-CoV-2 infection. Nevertheless, these findings are consistent with preclinical data indicating that the inhibition of the ASM/ceramide system by fluoxetine prevents infection of Vero E6 cells with pp-VSV-SARSCoV-2 spike [12] and that the reconstitution of ceramides in cells treated with fluoxetine restores the infection [12]. They are also in line with observational study results suggesting the utility of medications with high FIASMA activity against COVID-19 disease progression [23,24,36]. The modification of the metabolic ratio hexosylceramide/ceramide by fluoxetine is a novel finding of this study that might explain its observed antiviral and anti-inflammatory effects, as this ratio has been shown to correlate with clinical disease severity and inflammation markers in patients with COVID-19 [55]. This result is compatible with either increased activity of glucosylceramidases/galactosylceramidases (leading to the hydrolysis of hexosylceramides to ceramides) or decreased activity of glucosylceramide synthase/galactosylceramide synthase (converting ceramides to hexosylceramides) upon SARS-CoV-2 infection. Further research is required to delineate the explicative weight of these different hypotheses [55,56,57,58]. Altogether, these data suggest that the ceramide system could be a relevant treatment target in COVID-19, especially because via its mode of action, fluoxetine maintains activity against SARS-CoV-2 variants.

Importantly, because the blood half-life of fluoxetine is about 2 to 4 days in humans and experimental data showed a human fluoxetine concentration lung:plasma split of 60:1 [59], fluoxetine may lead to subsequent potential antiviral and anti-inflammatory effects in the lungs quickly after its dispensation. This assertion is supported by a population pharmacokinetic modelling study [60] predicting that most patients would reach the antiviral EC_90_ target, obtained from Calu-3 cells, in the lung tissues by day 1 at 40 mg/d and by day 3 at 20 mg/d. It is also supported by an experiment conducted in healthy volunteers showing that oral use of another FIASMA antidepressant, amitriptyline, whose blood half-life is about 20 h, prevents infection of freshly isolated nasal epithelial cells several hours after having taken the treatment, which is restored after the reconstitution of ceramides in these cells [12].

Fluoxetine is a prescription medication with a number of potential side effects and drug interactions, which are detailed in the Summary of Product Characteristic (SmPC) [61] and should be considered by the treating physician prior to use. Particularly, consistent with the medical contraindications, fluoxetine should not be used in combination with irreversible nonselective monoamine oxidase inhibitors or metoprolol. Fluoxetine is included in the World Health Organization’s Model (WHO) List of Essential Medicines [62]. When selecting essential medicines, their safety is also considered. Fluoxetine is the best tolerated antidepressant in the treatment of patients with depression; discontinuation rates are even lower than with placebo treatment [63]. In the U.S., over 10% of the adult population has taken an antidepressant in the past 30 days [64], with fluoxetine being one of the most commonly prescribed antidepressants [65]. This means that fluoxetine is used very widely without major problems.

To the best of our knowledge, this study is the first to identify fluoxetine as a potential anti-SARS-CoV-2 medication in an animal model, through mechanisms that result in both antiviral and anti-inflammatory properties. Since fluoxetine is easy to use as a once-a-day pill, shows high safety margins and good tolerability, is widely available at low cost, and has, among SSRIs, both the greatest in vitro inhibition effect on the ceramide system and antiviral effect [13,15,34], this molecule should be prioritized in large-scale clinical trials at different stages of the disease [6,10], either alone or in combination with other medications. Given the promising results of the FIASMA antidepressant fluvoxamine in four clinical trials [27,28,30,33], fluoxetine, if proven effective in clinical trials, would enrich the current therapeutic arsenal with an inexpensive, well-tolerated, and easily administered medication in the global fight against COVID-19.

## 4. Materials and Methods

### 4.1. Safety Procedures

All described experiments involving SARS-CoV-2 infections have been entirely performed and processed in a Biosafety Level 3 (BSL-3) facility. Prior samples from infected cells being taken out of the BSL-3 for subsequent analysis, and all SARS-CoV-2 inactivation procedures were validated and approved by Institute of Pharmacology and Structural Biology (IPBS) biosecurity committee.

### 4.2. SARS-CoV-2 Virus Production

Virus productions were performed as previously described [51]. Briefly, SARS-CoV-2 isolates were amplified by infecting Vero E6 cells (ATCC CRL-1586) (MOI 0.005) in DMEM (Gibco™, Thermo Fisher Scientific, Waltham, MA, USA) supplemented with 10 mM HEPES and 1% penicillin-streptomycin (Gibco™, Thermo Fisher Scientific, Waltham, MA, USA). The supernatant was harvested at 48 h post-infection when cytopathic effects were observed, cell debris were removed by centrifugation, and aliquots were frozen at −80 °C. Viral stocks were titrated by TCID50 assays in Vero E6 cells (see Appendix A).

### 4.3. Virus Titration by TCID_50_ Calculation

The day prior to infection, 50,000 VeroE6 cells per well were seeded in 96-well tissue culture plates using 10% FBS DMEM, and then incubated overnight at 37 °C in a humidified, 5% CO_2_ atmosphere-enriched chamber. Serial 2.5-fold dilutions (from 10^−1^ to 10^−6.5^) of the samples were prepared in DMEM and used to infect Vero E6 cells; each dilution was tested in four replicates. The plates were incubated for at least 96 h and observed to monitor the development of cytopathic effect using an EVOS Floid microscope (Invitrogen). Viral titres, expressed as median tissue culture infectious dose (TCID_50_/mL), were calculated according to both Reed and Muench and Karber methods based on three or four replicates for dilution [66].

### 4.4. Cell Infection and Treatment with Fluoxetine

For in vitro experiments, fluoxetine hydrochloride (Tocris Bio-Techne, Noyal Châtillon sur Seiche, France; CAS Number: 56296-78-7) was dissolved in H_2_O_2_ at 10 mM concentration and stored at −20 °C until use. The day prior to infection, 50,000 Vero E6 or A549-ACE2-TMPRSS2 cells were seeded in 96-well tissue culture plates in 10% FBS DMEM supplemented with 10 mM HEPES (Gibco™) and 1% penicillin-streptomycin (Gibco™), and then incubated overnight at 37 °C in a humidified, 5% CO_2_ atmosphere-enriched chamber. One day later, cells were treated with fluoxetine (12.5 µM) for 2 h prior infection, and then infected with SARS-CoV-2 strains at indicated MOI in 90 µL DMEM supplemented with 10 mM HEPES, 1% penicillin-streptomycin and 1% L-Glutamine for 1 h at 37 °C. Then, culture medium was completed with 90 µL of culture medium containing fluoxetine at 25 µM.

### 4.5. Air Liquid Interface (ALI) Cultures

Air-liquid interface cultures were established as previously described [52]. Briefly, to obtain ALI cultures, human airway organoids, derived as previously described [67,68], were dissociated into single cells using TrypLE express. Cells were seeded at a density of ~50,000 cells per transwell on human placenta collagen I-coated Transwell^®^ permeable supports (6.5 mm diameter; 0.4 µm pore size; Corning) and cultured with organoid medium [67,68]. The medium was replaced 48 h after seeding, and once the cells reached confluence (~3 days after seeding), the medium was removed from the apical side of the transwell and replaced by differentiation media (STEMCELL Technologies, Saint Égrève, France; PneumaCult™-ALI Medium (Catalog #05001)). The medium was refreshed every 3 days and mucus removed with warm PBS every week. After 3–4 weeks of differentiation, ALI were used for SARS-CoV-2 infection. Prior to infection, the apical side of ALI were washed with PBS to remove excess of mucus and then pre-treated with fluoxetine on apical (100 µL) and basolateral (500 µL) sides for 2 h at the indicated concentrations. Then ALI were infected with 10 µL of SARS-CoV-2 (5 × 10^5^ PFU/ALI) on the apical side for 4 h, washed with 150 µL of warmed PBS and then incubated for 72 h while maintaining fluoxetine on the basolateral side. Apical side of each ALI was washed with 150 µL of warm PBS to assess live viral load by TCID_50_ assay on Vero E6 cells. ALI were fixed with 4% paraformaldehyde overnight before being taken out of the BSL-3. For staining, cells were permeabilized with 0.2% Triton X-100, then membranes were excised in 2 mL Eppendorf tubes containing blocking buffer (PBS 3% BSA) and incubated for 20 min at room temperature. Primary antibodies were incubated overnight at 4 °C (Appendix A). After three washes with 0.2% Triton X-100 in PBS, secondary antibodies were incubated for 1 h at room temperature in the dark. Three washes were performed and following staining with DAPI. Slides were mounted in Vectafield Antifade Mounting Medium with coverslips on membranes. Images were acquired with an Olympus FV100 inverted confocal microscope.

### 4.6. Cell Death and Viability

Cell death and viability were measured 72 h after infection, as previously described [51] (see Appendix A).

### 4.7. Pharmacokinetics of Fluoxetine in Mice

Pharmacokinetic studies were approved by the Committee on the Ethics of Animal Experiments of the Government of Unterfranken, Germany (License 55.2-2532.1-27/11) and Landesamt für Natur, Umwelt und Verbraucherschutz Nordrhein-Westfalen, Germany (License 81-02.04.2021.A112) and performed according to the European Union animal protection directive (Directive 2010/63/EU).

C57BL/6 male mice (Charles River, Sulzfeld, Germany) received 5 mg/kg fluoxetine (fluoxetine hydrochloride; Sigma Aldrich, Darmstadt, Germany) orally and serum levels of fluoxetine and its active metabolite desmethylfluoxetine were measured over time as indicated (Appendix A). To determine the appropriate dose of fluoxetine that efficiently reduces the activity of ASM in the lung, mice were injected intraperitoneally (i.p.) with saline (0.9% NaCl solution) or different doses of fluoxetine as indicated. Blood and lung tissue were collected 6 h later. As shown in Appendix A, serum level of fluoxetine and its metabolite peaked at 6 h and showed a half-life around 20 h. Similar serum levels were obtained by i.p. administration of fluoxetine. ASM activity in lung lysates was assessed as previously described [69], and revealed that a dose of 40 mg/kg i.p. is required to significantly inhibit lysosomal ASM activity in the lung of fluoxetine-treated C57BL/6 mice (Appendix A). Therefore, we evaluated the efficacy of 40 mg/kg i.p. fluoxetine on SARS-CoV-2 infection in K18-hACE2.

### 4.8. Mouse Infection

Mouse infection studies were approved by local and national ethic committees (License APAFIS#27729-2020101616517580 v3, Minister of Research, France) and performed according to local guidelines (French ethical laws) and the European Union animal protection directive (Directive 2010/63/EU).

K18-hACE2 mice (Jackson Laboratory-USA, female, weight between 18 and 20 g, 8 (day 6) to 16 (day 2) per group and time point) were infected under a short anaesthesia (isoflurane 4%), by the intranasal route with 10^5^ PFU of SARS-CoV-2 (strain BetaCoV/France/IDF0372/2020). Thereafter, 4 h post-infection and then once daily, the first group received the vehicle (saline) treatment and the second group received fluoxetine (40 mg/kg i.p.) (Figure 1). During the 6 days of infection, mice were monitored daily for body weight as a measure of disease progression. At day 2 and 6 post-infection, after a terminal anaesthesia (ketamine 100 mg/kg + xylazine 10 mg/kg i.p.), the lung tissues were harvested and homogenated in gentleMACS™ M Tubes (Miltenyi Biotec, Paris, France) containing 3 mL of DPBS (Gibco™) using a gentleMACS™ Dissociator (Miltenyi Biotec, Paris, France). At day 6, body temperature was recorded using a lubricated rectal probe before mouse euthanasia.

### 4.9. Quantitative Real-Time PCR Analysis of Viral RNA and Inflammatory Genes

For each mouse, 150 µL of lung tissue homogenate was mixed with 1 mL TRIzol Reagent (Invitrogen) and stored at −80 °C at least 48 h before being taken out the BSL-3 facility. Total RNA was extracted, reverse-transcribed and gene expression levels of cytokines and chemokines were assessed as previously described [67]. Viral loads were performed as previously described [70,71]. Briefly, viral load quantifications were carried out by linear regression employing a standard curve of six known quantities of plasmids containing the RdRp sequence (10^7^ to 100 copies) and qPCRs were performed in TaqMan Universal PCR Master Mix (Thermo Fisher Scientific). qPCRs for gene expression and viral load were performed in triplicate and assessed with an ABI 7500 real-time PCR system (Applied Biosystems, Thermo Fisher Scientific, Waltham, MA, USA). Primers and probe sequences are provided in Appendix A.

### 4.10. Sphingolipid Quantification by Liquid Chromatography Tandem-Mass Spectrometry (LC-MS/MS)

Sphingolipids were extracted from lung tissue homogenates (20 µL) as previously described [72]. To this end 1.5 mL methanol/chloroform (2:1, *v:v*) containing the internal standards 17:0 ceramide (d18:1/17:0), d_31_-16:0 sphingomyelin (d18:1/16:0-d_31_) and 17:0 glucosyl(β) ceramide (d18:1/17:0) (all Avanti Polar Lipids, Alabaster, AL, USA) were added. Extraction was facilitated by incubation at 48 °C with gentle shaking (120 rpm) overnight. To reduce interference from glycerolipids, samples were saponified with 150 μL 1 M methanolic KOH for 2 h at 37 °C with gentle shaking (120 rpm) followed by neutralization with 12 μL glacial acetic acid. After centrifugation at 2200 g for 10 min at 4 °C, organic supernatants were evaporated to dryness using a Savant SpeedVac concentrator (Thermo Fisher Scientific, Dreieich, Germany). Dried residues were reconstituted in 200 μL acetonitrile/methanol/water (47.5:47.5:5 (*v:v:v*), 0.1% formic acid) and subjected to LC-MS/MS sphingolipid quantification applying the multiple reaction monitoring (MRM) approach. Chromatographic separation was achieved on a 1290 Infinity II HPLC (Agilent Technologies, Waldbronn, Germany) equipped with a Poroshell 120 EC-C8 column (3.0 × 150 mm, 2.7 μm; Agilent Technologies) guarded by a pre-column (3.0 × 5 mm, 2.7 μm) of identical material. MS/MS analyses were carried out using a 6495 triple-quadrupole mass spectrometer (Agilent Technologies) operating in the positive electrospray ionization mode (ESI+). Chromatographic conditions and settings of the ESI source and MS/MS detector have been published elsewhere [73]. Peak areas of Cer 16:0 (*m/z* 520.5 → 264.3), SM 16:0 (*m/z* 703.6 → 184.1), and HexCer 16:0 (*m/z* 700.6 → 264.3) subspecies, as determined with MassHunter software (version 10.1, Agilent Technologies), were normalized to those of their internal standards. Quantification was performed with MassHunter Software (version 10.1, Agilent Technologies). Further, the determined Cer, SM and HexCer amounts were normalized to the actual protein content of the tissue homogenate used for extraction determined by the Coomassie (Bradford) Protein Assay (Thermo Fisher Scientific, Waltham, MA, USA).

### 4.11. Dual A549 Cell Stimulation

To investigate whether fluoxetine modulates NF-kB and interferon pathways, we used A549–Dual cells carrying both an NF-κB-driven secreted phosphatase alkaline (SEAP) gene and an IRF-driven Luciferase reporter genes (InvivoGen, Toulouse, France). Cells were plated in 96-well plates at 5 × 10^4^ cells per well and incubated at 37 °C. The following day, cells were pre-incubated with or without fluoxetine (12.5–25 µM) for 1 h before treatment with various activators of the NF-kB and/or IRF pathways including TNF-α (10–0.1 ng/mL), IFN-α (10–0.1 ng/mL), and poly I::C (10–0.1 µg/mL, InvivoGen, Toulouse, France). After 24 h of incubation, supernatant was collected and the presence of SEAP and Lucia reporting respectively for NF-kB and IRF activation was quantified following incubation with SEAP and Lucia enzyme substrates Quanti-blue and Quanti-luc, respectively, according to the manufacturer’s instructions (InvivoGen, Toulouse, France).

### 4.12. Statistical Analysis

All data in the text and figures are expressed as mean (±SEM). All means were compared using *t*-tests or Mann-Whitney tests when assumptions were not met. For all associations, we performed residual analyses to assess the fit of the data, checked assumptions, and examined the potential influence of outliers. Outliers were defined as having values outside the 1.5 interquartile range (IQR). Analyses were carried out with GraphPad Prism 9 and R Software, version 4.1.3. Statistical significance was considered when two-sided *p* < 0.05.

## Figures and Tables

**Figure 1 ijms-23-13623-f001:**
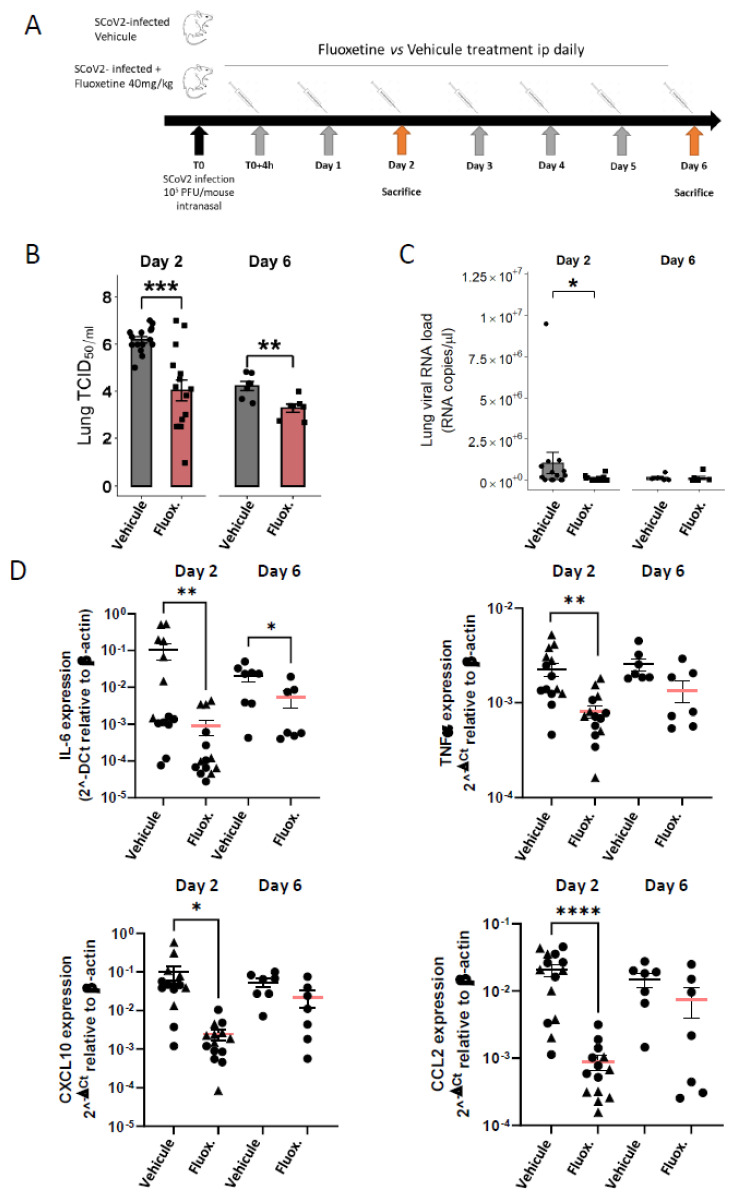
Fluoxetine reduced viral load and inflammatory response in K18-hACE2 mice infected with SARS-CoV-2. (**A**) Experimental design of the study; (**B**) infectious viral loads in the lungs of vehicle- and fluoxetine-treated mice infected with SARS-CoV-2 at day 2 and 6 post-infection expressed as log10 TCID50/mL; (**C**) viral RNA levels in the lungs of vehicle- and fluoxetine-treated mice infected with SARS-CoV-2 at day 2 and 6 post-infection expressed as Lung SARS-CoV-2 RNA load (RNA copies/µL). (**D**) Cytokine and chemokine transcripts in lung tissue at day 2 and 6 post-infection of SARS-CoV-2-infected mice, treated with vehicle or with fluoxetine. *, *p* < 0.05; **, *p* < 0.01; ***, *p* < 0.001; ****, *p* < 0.0001.

**Figure 2 ijms-23-13623-f002:**
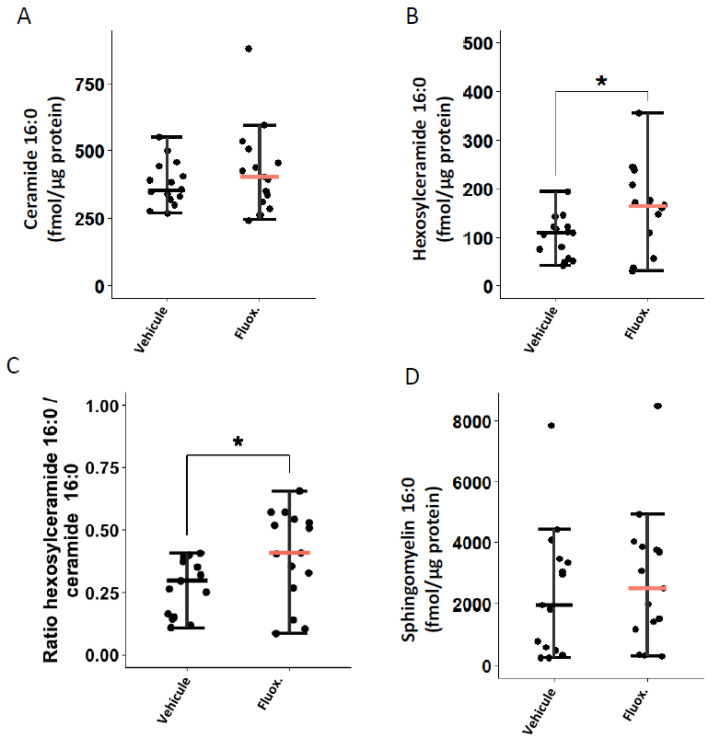
Fluoxetine effect on lung levels of (**A**) ceramide 16:0, (**B**) hexosylceramide 16:0, (**C**) hexosylceramide 16:0 / ceramide 16:0 ratio, and (**D**) sphingomyelin 16:0, at day 2 post-infection. *, *p* < 0.05.

**Figure 3 ijms-23-13623-f003:**
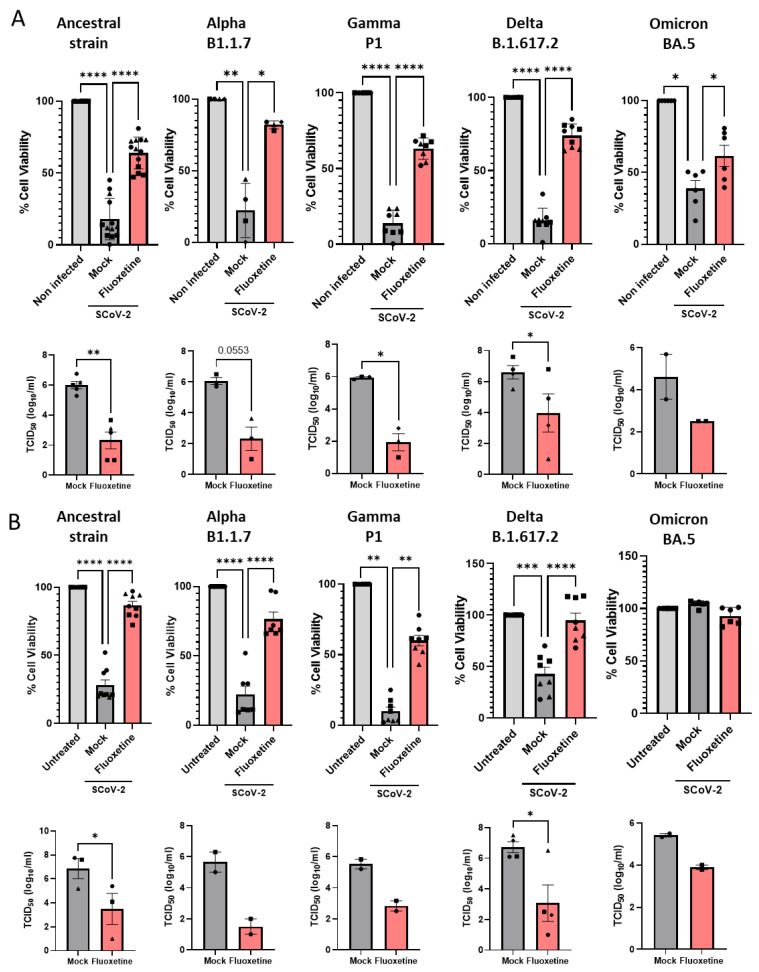
Fluoxetine activity on SARS-CoV-2 variants in vitro. (**A**) Activity of fluoxetine (12.5 µM) or mock (water) in Vero E6 cells infected with SARS-CoV-2 (Pasteur, B.1), B.1.1.7 (UK), P1, B.1.617 (Delta) and BA.5 (Omicron) variants measured by assessing cell viability and viral load. (**B**) Antiviral activity of fluoxetine (12.5 µM) measured in A549 cells expressing hACE2-TMPRSS2 infected with SARS-CoV-2 (Pasteur, B.1), B.1.1.7 (UK), P1, B.1.617 (Delta) and BA.5 (Omicron) variants measured by cell viability and viral load. Representative of two to seven independent experiments, data are expressed as mean + SEM. *, *p* < 0.05, **, *p* < 0.01; ***, *p* < 0.001; ****, *p* < 0.0001.

**Figure 4 ijms-23-13623-f004:**
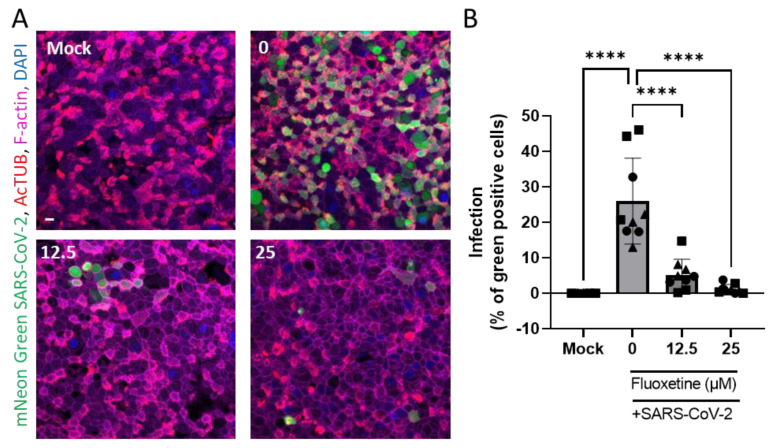
Fluoxetine inhibits SARS-CoV-2 infection of 2D human airway cells. (**A**) Representative images of immunofluorescent stainings of mNeon Green SARS-CoV-2 infected airway cultures. Acetylated tubulin (AcTUB, red) stains cell ciliates, phalloidin-coupled Alexa Fluor 647 stains F-actin (purple) and DAPI stains nuclei (blue). Scale bar = 10 µm. (**B**) Live virus titre measured on apical washes 72 h after infection with mNeon Green SARS-Cov-2. Data from three independent experiments on two independent donors, expressed as mean ± SEM, **** *p* < 0.0001.

## Data Availability

Data available in a publicly accessible repository. The data presented in this study are openly available in FigShare at doi:10.6084/m9.figshare.21460026.

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
