# Peer review of "Antiviral and Anti-Inflammatory Activities of Fluoxetine in a SARS-CoV-2 Infection Mouse Model"

_ijms, 2022, doi:10.3390/ijms232113623_

Round 1
Reviewer 1 Report
The present manuscipt can be published in the present form.
Author Response
- Comments and Suggestions for Authors :
1/ The present manuscipt can be published in the present form.
The authors thank the reviewer for his/her positive and fair evaluation of the manuscript.
Of note, during the revision process of the manuscript, we had access to human airway epithelial cells cultured as Air Liquid Interface. As shown in Revised Main Figure 4, fluoxetine treatment inhibited SARS-CoV-2 infection reflected by the reduction in the % of Neon Green positive cells, confirming fluoxetine effect on SARS-CoV-2 infection in human primary epithelial cells.

Reviewer 2 Report
In the manuscript by Pericat et al. the authors demonstrate an antiviral action of Fluoxetine against SARS CoV-2 virus in the mouse infection model.
Although there are several publications demonstrating the anti-SARS-CoV-2 action of fluoxetine and related compounds in vitro and there are some retrospective and prospective studies in humans suggesting an anti-Covid 19 effect, the manuscript still contributes new and important data since it demonstrates action in the mouse model which bridges an existing gap of knowledge. Nevertheless there are still some points that need to be addressed.
Major points:
The authors statement that inhibition of cytokines is uncoupled from inhibition of virus titers is not convincingly shown by the data. The problem here is, that the two parameters are measured in the infection scenario and thus reduced titers would always lead to a reduction in cytokine expression because of decreased levels of PAMPs. The authors should seek for a method to analyse influence on cytokine expression in the absence of a dynamic stimulus such as infection (e.g. vRNA stimulation, infection with inactivated viruses). Furthermore, to justify that there is an independent immunemodulatory action, more readouts such as only a few cytokines would be needed.
While the authors showed reduction of titers of several variants, the currently most important variant omicron is not included and should also be tested, at least in vitro. This is important because omicron is supposed to use a different entry mechanism compared to beta or delta.
The effects on the ratio of HexCer/Cer are poor. Since the authors state that this a major novel finding of the manuscript they should either seek for an additional method to more convincingly demonstrate the modulation of the ceramide system or tune the statement down, that this is the functional cause of virus inhibition.
Since effects on cell viability are an important issue when using drugs, it is not sufficient to test cell viability only in the LDH assay. Additional methods such as MTT should be also employed.
Minor points:
There are still some orthographical errors that should be corrected.
Fluoxetine and related compounds are antidepressants, the authors need to discuss potential side effects, especially in unstable people when broadly used as an antiviral drug.
Author Response
In the manuscript by Pericat et al. the authors demonstrate an antiviral action of Fluoxetine against SARS CoV-2 virus in the mouse infection model.
Although there are several publications demonstrating the anti-SARS-CoV-2 action of fluoxetine and related compounds in vitro and there are some retrospective and prospective studies in humans suggesting an anti-Covid 19 effect, the manuscript still contributes new and important data since it demonstrates action in the mouse model which bridges an existing gap of knowledge. Nevertheless there are still some points that need to be addressed.
Major points:
1/ The authors statement that inhibition of cytokines is uncoupled from inhibition of virus titers is not convincingly shown by the data. The problem here is, that the two parameters are measured in the infection scenario and thus reduced titers would always lead to a reduction in cytokine expression because of decreased levels of PAMPs. The authors should seek for a method to analyse influence on cytokine expression in the absence of a dynamic stimulus such as infection (e.g. vRNA stimulation, infection with inactivated viruses). Furthermore, to justify that there is an independent immunemodulatory action, more readouts such as only a few cytokines would be needed.
To address the comment raised by the reviewer, we assessed global the NF-κB and the Interferon Regulatory Factor (IRF) pathways modulation by fluoxetine using the A549-Dual™ cells consisting in a human NF-κB-SEAP & IRF-Luc Reporter lung carcinoma cell line. Cells were stimulated with PolyIC to mimic vRNA stimulation, and TNFa or IFNa as a positive controls. As shown in SuppFigure 3D-E, fluoxetine inhibits NF-κB activation, while IRF pathway was not affected. These results are now included in the Revised Manuscript (page 5, line 174).
2/ While the authors showed reduction of titers of several variants, the currently most important variant omicron is not included and should also be tested, at least in vitro. This is important because omicron is supposed to use a different entry mechanism compared to beta or delta.
As suggested by the reviewer, we extended our analysis on the BA.5 Omicron variant. As described for the other variants, fluoxetine showed reduced titer of the omicron variant associated with restoration of cell viability of Ver E6 cells. These data are now included in the Revised Manuscript (page 7, line 209) and in the Revised Figure 3A-B. Interestingly, while Omicron replicates in A549 expressing hACE2 and TMPRSS2, infection with this variant does not result in cytopathic effect after 72h of infection, which might be due to a lower infectivity in this cell type. Investigation of the potential mechanisms of action is out of the scope of this manuscript and will require further investigation.
3/ The effects on the ratio of HexCer/Cer are poor. Since the authors state that this a major novel finding of the manuscript they should either seek for an additional method to more convincingly demonstrate the modulation of the ceramide system or tune the statement down, that this is the functional cause of virus inhibition.
We agree with the reviewer that at this stage of the study, the effects on the ratio of HexCer/Cer is correlative with the antiviral effect of fluoxetine. As suggested, we tuned down our statement (see Revised Manuscript page 9, line 267).
4/ Since effects on cell viability are an important issue when using drugs, it is not sufficient to test cell viability only in the LDH assay. Additional methods such as MTT should be also employed.
We agree with the reviewer that the drug effect must be evaluated by independent read-outs. To access fluoxetine effect on SARS-CoV-2 infection, we used 3 independent read-outs, first fluoxetine effect on viral load using the TCID50 assay, and on virus-induced cytotoxicity by measuring in parallel cell death (LDH release in cell supernatant, Takara) and cell viability by quantifying cellular ATP level (Cell Titer Glo 2.0, Promega). Suggested by the reviewer, MTT is a cell viability test based on tetrazolium cleavage by mitochondrial dehydrogenases of viable cells yielding purple formazan crystals. Instead of MTT test, for cell viability evaluation, we used the CellTiter-Glo® 2.0 Assay that determines the number of viable cells in culture by quantifying ATP, indicative of the presence of metabolically active cells. Considering that MTT test and Cell Titer Glo both evaluate cell viability, we present in Revised Figure 3 and Revised Supplemental Figure 4 results of the effect of fluoxetine on SARS-CoV-2 infection by combining measures of viral load, cell viability and cell death on two different cell types and five viral strains.
Minor points:
5/ There are still some orthographical errors that should be corrected.
The manuscript has been carefully read to correct the orthographical errors.
6/ Fluoxetine and related compounds are antidepressants, the authors need to discuss potential side effects, especially in unstable people when broadly used as an antiviral drug.
Thanks to the reviewer comment, we discussed fluoxetine safety and its management of potential side effect in the discussion of the Revised Manuscript (page 9, line 299).
Of note, during the revision process of the manuscript, we had access to human airway epithelial cells cultured as Air Liquid Interface. As shown in Revised Main Figure 4, fluoxetine treatment inhibited SARS-CoV-2 infection reflected by the reduction in the % of Neon Green positive cells, confirming fluoxetine effect on SARS-CoV-2 infection in human primary epithelial cells.
In the manuscript by Pericat et al. the authors demonstrate an antiviral action of Fluoxetine against SARS CoV-2 virus in the mouse infection model.
Although there are several publications demonstrating the anti-SARS-CoV-2 action of fluoxetine and related compounds in vitro and there are some retrospective and prospective studies in humans suggesting an anti-Covid 19 effect, the manuscript still contributes new and important data since it demonstrates action in the mouse model which bridges an existing gap of knowledge. Nevertheless there are still some points that need to be addressed.
Major points:
1/ The authors statement that inhibition of cytokines is uncoupled from inhibition of virus titers is not convincingly shown by the data. The problem here is, that the two parameters are measured in the infection scenario and thus reduced titers would always lead to a reduction in cytokine expression because of decreased levels of PAMPs. The authors should seek for a method to analyse influence on cytokine expression in the absence of a dynamic stimulus such as infection (e.g. vRNA stimulation, infection with inactivated viruses). Furthermore, to justify that there is an independent immunemodulatory action, more readouts such as only a few cytokines would be needed.
To address the comment raised by the reviewer, we assessed global the NF-κB and the Interferon Regulatory Factor (IRF) pathways modulation by fluoxetine using the A549-Dual™ cells consisting in a human NF-κB-SEAP & IRF-Luc Reporter lung carcinoma cell line. Cells were stimulated with PolyIC to mimic vRNA stimulation, and TNFa or IFNa as a positive controls. As shown in SuppFigure 3D-E, fluoxetine inhibits NF-κB activation, while IRF pathway was not affected. These results are now included in the Revised Manuscript (page 5, line 174).
2/ While the authors showed reduction of titers of several variants, the currently most important variant omicron is not included and should also be tested, at least in vitro. This is important because omicron is supposed to use a different entry mechanism compared to beta or delta.
As suggested by the reviewer, we extended our analysis on the BA.5 Omicron variant. As described for the other variants, fluoxetine showed reduced titer of the omicron variant associated with restoration of cell viability of Ver E6 cells. These data are now included in the Revised Manuscript (page 7, line 209) and in the Revised Figure 3A-B. Interestingly, while Omicron replicates in A549 expressing hACE2 and TMPRSS2, infection with this variant does not result in cytopathic effect after 72h of infection, which might be due to a lower infectivity in this cell type. Investigation of the potential mechanisms of action is out of the scope of this manuscript and will require further investigation.
3/ The effects on the ratio of HexCer/Cer are poor. Since the authors state that this a major novel finding of the manuscript they should either seek for an additional method to more convincingly demonstrate the modulation of the ceramide system or tune the statement down, that this is the functional cause of virus inhibition.
We agree with the reviewer that at this stage of the study, the effects on the ratio of HexCer/Cer is correlative with the antiviral effect of fluoxetine. As suggested, we tuned down our statement (see Revised Manuscript page 9, line 267).
4/ Since effects on cell viability are an important issue when using drugs, it is not sufficient to test cell viability only in the LDH assay. Additional methods such as MTT should be also employed.
We agree with the reviewer that the drug effect must be evaluated by independent read-outs. To access fluoxetine effect on SARS-CoV-2 infection, we used 3 independent read-outs, first fluoxetine effect on viral load using the TCID50 assay, and on virus-induced cytotoxicity by measuring in parallel cell death (LDH release in cell supernatant, Takara) and cell viability by quantifying cellular ATP level (Cell Titer Glo 2.0, Promega). Suggested by the reviewer, MTT is a cell viability test based on tetrazolium cleavage by mitochondrial dehydrogenases of viable cells yielding purple formazan crystals. Instead of MTT test, for cell viability evaluation, we used the CellTiter-Glo® 2.0 Assay that determines the number of viable cells in culture by quantifying ATP, indicative of the presence of metabolically active cells. Considering that MTT test and Cell Titer Glo both evaluate cell viability, we present in Revised Figure 3 and Revised Supplemental Figure 4 results of the effect of fluoxetine on SARS-CoV-2 infection by combining measures of viral load, cell viability and cell death on two different cell types and five viral strains.
Minor points:
5/ There are still some orthographical errors that should be corrected.
The manuscript has been carefully read to correct the orthographical errors.
6/ Fluoxetine and related compounds are antidepressants, the authors need to discuss potential side effects, especially in unstable people when broadly used as an antiviral drug.
Thanks to the reviewer comment, we discussed fluoxetine safety and its management of potential side effect in the discussion of the Revised Manuscript (page 9, line 299).
Of note, during the revision process of the manuscript, we had access to human airway epithelial cells cultured as Air Liquid Interface. As shown in Revised Main Figure 4, fluoxetine treatment inhibited SARS-CoV-2 infection reflected by the reduction in the % of Neon Green positive cells, confirming fluoxetine effect on SARS-CoV-2 infection in human primary epithelial cells.

Reviewer 3 Report
1. Remove the word Mann Whitney ttest use only p value.
2. The authors must conduct relevant experiments o prove the involvement of ceramide system
3. In 4.2. SARS-CoV-2 virus production
Virus productions were performed as previously described41.- Briefly describe the method.
Author Response
- Remove the word Mann Whitney ttest use only p value.
As requested by the reviewer, Mann Whitney ttest words have been removed from the text.
- The authors must conduct relevant experiments to prove the involvement of ceramide system
We agree with the reviewer that at this stage of the study, the effects on the ratio of HexCer/Cer is correlative with the antiviral effect of fluoxetine. Due to the fine-tuned regulation of the ceramide system, we tuned down the message on the involvement of this pathway in the fluoxetine effect on SARS-CoV-2 infection. In the context of this revision, we believe that the extended experiments required to fully characterize how the ceramide pathway might be implicated in fluoxetine effect are beyond the scope of this study. We have tuned our statement down (see Revised Manuscript page 9, line 267).
- In 4.2. SARS-CoV-2 virus production
Virus productions were performed as previously described41.- Briefly describe the method.
As requested by the reviewer, the method in section 4.1 is now briefly described in the Revised Manuscript (Page 11, line 329).
Of note, during the revision process of the manuscript, we had access to human airway epithelial cells cultured as Air Liquid Interface. As shown in Revised Main Figure 4, fluoxetine treatment inhibited SARS-CoV-2 infection reflected by the reduction in the % of Neon Green positive cells, confirming fluoxetine effect on SARS-CoV-2 infection in human primary epithelial cells.
